# Lipophilic Toxins in Wild Bivalves from the Southern Gulf of California, Mexico

**DOI:** 10.3390/md19020099

**Published:** 2021-02-09

**Authors:** Ignacio Leyva-Valencia, Jesús Ernestina Hernández-Castro, Christine J. Band-Schmidt, Andrew D. Turner, Alison O’Neill, Erick J. Núñez-Vázquez, David J. López-Cortés, José J. Bustillos-Guzmán, Francisco E. Hernández-Sandoval

**Affiliations:** 1CONACYT-Instituto Politécnico Nacional, Centro Interdisciplinario de Ciencias Marinas, La Paz, B.C.S. 23096, Mexico; 2Instituto Politécnico Nacional, Centro Interdisciplinario de Ciencias Marinas, La Paz, B.C.S. 23096, Mexico; ernestina_0905@hotmail.com (J.E.H.-C.); cjbands@ipn.mx (C.J.B.-S.); 3The Centre for Environment, Fisheries and Aquaculture Science, Weymouth, Dorset DT4 8UB, UK; andrew.turner@cefas.co.uk (A.D.T.); alison.oneill@cefas.co.uk (A.O.); 4Centro de Investigaciones Biológicas del Noroeste, La Paz, B.C.S. 23096, Mexico; enunez04@cibnor.mx (E.J.N.-V.); jose04@cibnor.mx (J.J.B.-G.); fhernan04@cibnor.mx (F.E.H.-S.)

**Keywords:** lipophilic toxins, dinoflagellates, cyclic imines, bivalves, LC-MS/MS

## Abstract

Most of the shellfish fisheries of Mexico occur in the Gulf of California. In this region, known for its high primary productivity, blooms of diatoms and dinoflagellates are common, occurring mainly during upwelling events. Dinoflagellates that produce lipophilic toxins are present, where some outbreaks related to okadaic acid and dinophisystoxins have been recorded. From January 2015 to November 2017 samples of three species of wild bivalve mollusks were collected monthly in five sites in the southern region of Bahía de La Paz. Pooled tissue extracts were analyzed using LC-MS/MS to detect lipophilic toxins. Eighteen analogs of seven toxin groups, including cyclic imines were identified, fortunately individual toxins did not exceed regulatory levels and also the total toxin concentration for each bivalve species was lower than the maximum permitted level for human consumption. Interspecific differences in toxin number and concentration were observed in three species of bivalves even when the samples were collected at the same site. Okadaic acid was detected in low concentrations, while yessotoxins and gymnodimines had the highest concentrations in bivalve tissues. Although in low quantities, the presence of cyclic imines and other lipophilic toxins in bivalves from the southern Gulf of California was constant.

## 1. Introduction

The Gulf of California (GuC) has been recognized for supporting extraordinary biological diversity, exceptionally primary high productivity and large populations of marine taxa of vertebrates and invertebrates [1]. The mollusk fisheries from the Gulf of California contribute to nearly 90% of the total production of Mexico [2]. In the northern GuC, *Panopea globosa* (geoduck clam) is an important exportation product for the Asian market, while in southern GuC, *Megapitaria squalida* (chocolate clam), *Dosinia ponderosa* (white clam), *Atrina maura* (pen shell) and other bivalves, are important components of the regional gastronomic culture. Most of these species are harvested from wild populations, where their growth is sustained by feeding on the natural phytoplankton community. Consequently, they can accumulate toxins produced by diatoms and dinoflagellates, even without an evident harmful algal bloom. For this reason, bivalves are the first vector of toxins towards humans.

Mollusks have limited mobility and they need to pump water to perform gas exchange and concentrate phytoplankton cells, their main food source to carry out their metabolic functions. Consequently, bivalves accumulate phycotoxins in distinct organs and tissues [3,4,5,6]. Distinct shellfish toxin syndromes such as paralytic shellfish poisoning (PSP), neurotoxic shellfish poisoning (NSP), amnesic shellfish poisoning (ASP) and diarrheic shellfish poisoning (DSP), have been described and associated with negative effects on human health, after shellfish consumption [7]. Acute effects in humans are well known [8,9], however the consequences resulting from chronic exposure are less well described. Each type of poisoning is related to a specific group of biotoxins formed naturally by distinct species of diatoms and dinoflagellates [9]. Toxins, can be classified according to their chemical structure as—paralytic shellfish toxins (PSTs) based on their parent molecule the saxitoxin (STX), amnesic shellfish toxins resulting from domoic acid (DA), neurotoxic toxins relating to brevetoxins (PbTXs), diarrheic toxins as okadaic acid (OA) and dinophysis toxins (DTXs), as well as additional lipophilic toxins (LT) including pectentoxins (PTXs), azaspiracids (AZAs), yessotoxins (YTXs) and cyclic imines (CI) [10].

Within the GuC region, species of the genera *Gymnodinium*, *Alexandrium* and *Pyrodinium* are known to produce PSTs, with *Gymnodinium catenatum* being the most studied dinoflagellate in the region [11]. The first scientific report of *G. catenatum* bloom of this species occurred in the northern GuC in 1939 [12,13], while the first PSP outbreak linked to this species extended from the coasts of Sonora (central GuC) to the coasts of Jalisco (Pacific Ocean) in 1979 [14]. Again in 2015, an extended bloom of this species was reported in northern GuC, triggering human poisoning, the death of marine organisms, the closure of fishing activities and economic losses in the region [15]. Other dinoflagellates, of the genera *Prorocentrum*, *Dinophysis* and *Phalacroma*, also inhabit the GuC. Some of their species can produce OA and other LT [16], while dinoflagellates such as *Gonyaulax spinifera*, *Protoceratium reticulatum*, *Azadinium spinosum* and *Vulcanodinium rugosum*, have been reported in the Pacific coast and in the GuC and are recognized to produce YTXs, AZAs and CI [17,18]. Epibenthic dinoflagellates, of the genus *Prorocentrum*, *Coolia*, *Ostreopsis*, *Amphidinium* and *Fukuyoa,* have been reported in Mexico. Usually, benthic dinoflagellates do not produce visible blooms, however they can originate dense cell aggregations known as benthic harmful algal blooms [19].

Only a few reports of LT-related outbreaks have been confirmed in Mexico, mainly in Bahía Todos Santos on the Pacific coastline and El Pardito island in the GuC [16,20]. The symptoms of DSP can be easily confused with bacterial or viral gastroenteritis, making accurate diagnosis of DSP outbreaks very difficult. Consequently, reliable records of DSP episodes do not exist, since acute symptoms are not always severe, affected people usually do not seek medical treatment, and/or doctors fail to identify the origin of the illness [21,22]. 

In Mexico, only OA, DTX and analogs, are included in the sanitary regulation (NOM-242-SSA1-2009), with an action limit of 160 µg/Kg. Yessotoxins and azaspiracids were incorporated in the Technical Guide of the Mexican Program of bivalve’s mollusks (Programa Mexicano de Sanidad en Moluscos Bivalvos, COFEPRIS), suggesting limits of 1 mg/Kg and 160 µg/Kg, respectively [23,24]. OA, which is the main diarrheic toxin, acts as a potent inhibitor of phosphatase activity in the cell membrane, mainly protein phosphatase 2A (PP2A) [25]. YTX causes cardiac damage in mice muscles and subsequently death after intraperitoneal injection, although their ecological role is unknown and there is no record of human intoxications [26,27]. AZAs are polyether toxins that produce similar symptoms to DSP, such as nausea, vomiting, severe diarrhea and stomach cramps, after eating mussels, intraperitoneal injection in mice caused neurotoxic effects and death within 20–90 min [28,29,30,31]. PTXs are a group of macrocyclic polyethers highly hepatotoxic to mice by intraperitoneal injection, however there is no evidence that PTXs have caused toxic effects in humans and probably diarrheic illness associated at these toxins are attributable to okadaic acid esters [32,33,34]. Cyclic imines were found in bivalve tissues in the early 1990s [35,36,37,38] and due to their high acute toxicity in mouse assays, are known as fast action toxins that may interfere with mouse bioassay to detect OA, brevetoxin (PbTX), YTX and AZA [38]. CI include: gymnodimines (GYM), spirolides (SPX), pinnatoxins (PnTX), prorocentrolides, spiro-prorocentrimines and portimines [35,36,37,38,39,40,41]. These toxins are macrocyclic compounds with an imine functionality (carbon-nitrogen double bond) and spiro-linked ether moieties, with the main molecular targets being muscle-type and neuronal nicotinic acetylcholine receptors (nAChR) [42,43]. Information regarding the presence of CI in shellfish from Mexico is scarce and the potential contamination of seafood products is difficult to assess. The quantification and structural confirmation have been compromised by the lack of toxin standards.

A range of biological, chemical and biomolecular assays have been developed for detection of LT [44]. In Mexico, the mouse bioassay (MBA), a commercial rapid test kit based on PP2A inhibition and LC-MS/MS are methods accepted for the surveillance and testing of DSP [45]. An advantage of the MBA, is the possibility to detect physiological and behavioral responses that cannot be detected using analytical methods. Conversely, if lower concentrations of OA-group toxins are combined with other LT, they can induce sub lethal and lethal effects in the mouse, resulting in false positives for DSP. Moreover, due to their poor specificity, MBA is not an appropriate method to detect CI [44].

To determine if samples of *M. squalida* (*n* = 5), *D. ponderosa* (*n* = 5) and *A. maura* (*n* = 3), collected in January and February 2015 contained LT, an exploratory mouse bioassay was performed, using mice males, strain CD-1 and weight of 18–22 g (Harlan Laboratories Ltd.). Extracts of LT following method described by Yasumoto and collaborators in 1978 [46], were injected intraperitoneally into each of three mice. In addition to diarrhea and lethargy, neurotoxicity signs as: hyperextension of the back, piloerection, tremors progressing to spam, stiffening and arching of the tail toward the head, paralysis and extension of the hind limbs, tremors of the whole body and respiratory arrest, suggested the presence of “fast action toxins,” as was described by Molgó and collaborators in 2014 and Hernández et al. 2017 [47,48]. Based on these analyses, the presence of LT was subsequently monitored using an analytical method (LC-MS/MS), which are described in this study.

The goal of this study was to determine the LT occurrence in three bivalve species used for human consumption in Bahía de La Paz, southern GuC, Mexico, to help mitigate against human health threats and to provide useful information for risk assessment of the management processes by future sanitary certifications.

## 2. Results and Discussion

### 2.1. Study Area and Sampling

The GuC is known for its exceptionally high nutrient concentrations that support a high primary productivity, sustaining the most important shellfish fisheries of Mexico for national and international markets [2,49]. A total of 420 shellfish samples consisting of white clam (*Dosinia ponderosa; n* = 68), chocolate clam (*Megapitaria squalida*; *n* = 104) and pen shell (*Atrina maura; n* = 248) were collected monthly between January 2015 and November 2017. Differences in inter-species sample numbers were due to variable environmental conditions (strong wind and scarce visibility under water) that did not allow field sampling. Differences in the substrate type influenced the distribution of these species in the Bay. White clam and chocolate clams were found buried within sand substrates and co-occurring in the same areas (sites S1, S2 and S5), whilst pen shells were semi-buried in muddy-sand substrates where they are found byssally attached (sites S3 and S4, Figure 1). 

In *Atrina maura* the more concentrated toxins were PnTXs (11.1 μg/Kg), where 10.1 μg/Kg corresponded to PnTXG. This concentration is lower than mouse LD_50_ of 45 μg/Kg reported for this CI [50]. However, lethal and sub lethal results observed by Hernández-Castro [48], could be due to combined effect of PnTX with other LTs, which were detected when LC-MS/MS analysis was performed.

Blooms of diatoms and dinoflagellates are common in southern GuC, mainly when upwelling events occur [17,51]. In contrast, studies of harmful algae in the GuC have focused on paralytic toxins and their producers [52,53,54,55,56]. To date, there are only two publications related to LT producers in this region [15,16]. In this research, cell densities < 1000 cells L^−1^ of DSP producers of the genera *Dinophysis* spp. and *Prorocentrum* spp. were observed in water samples collected in this study. Although blooms of these dinoflagellates were not detected [48], a bloom of the PST producer *G. catenatum*, occurred during June and July 2017, causing fishery and commercial closures established by the sanitary authorities. 

### 2.2. Content of Lipophilic Toxins in Three Species of Shellfish

The monthly monitoring of bivalves from 2015 to 2017 (25 months) from southern GuC, indicated a constant presence of eighteen analogs of LTs: Okadaic acid (OA), dinophysistoxin 1 and 2 (DTX1 and DTX2), pectenotoxins 1, 2 and 11 (PTX1, PTX2, PTX11), azaspiracids 1-3 (AZA1, AZA2, AZA3), yessotoxin (YTX), homo-yessotoxin (hYTX), 45-hydroxy yessotoxin (45 OH YTX), 45-hydroxy homo-yessotoxin (45 OH hYTX), 13-desmethyl spirolide C (SPX1), gymnodimine A (GYM) and the pinnatoxins E, F and G (PnTxE, PnTxF and PnTxG). These consequently represent seven main groups of LTs; OA-group including PTXs, YTXs, AZAs and cyclic imines (SPXs, GYM and PnTXs), Figure 2). The sum of concentrations for each group is shown in Table 1, Table 2 and Table 3. LC-MS/MS chromatograms obtained following acquisition using Multiple Reaction Monitoring (MRM) are illustrated in Appendix A.

In terms of food safety risks, the diarrheic toxins OA and DTXs were < 10% of the regulatory level for human consumption established by the National Sanitary Authority (COFEPRIS) [23,24]. Other toxin groups not regulated in Mexico, such as AZAs and SPXs, were found in low but detectable quantities, while the PnTXs, GYM and YTXs were the most concentrated in the analyzed samples. 

In water samples analyzed frequently, *Prorocentrum lima*, *P. rhathymum and Dinophysis caudata*, producers of OA [57,58] were found. In addition to the presence of *D. caudata*, PTX detected in mollusks samples can be explained by the presence of *D. acuta* and *Phalacroma rotundata* [59]. Other dinoflagellates recurrently observed were *Prorocentrum micans*, *P. koreanum and P. gracile*; however, the toxigenic potential of these species is unknown. *Protoceratium reticulatum* and *Gonyaulax spinifera* that produce YTXs, were observed in low cell densities [48]. *Azadinium spinosum, Alexandrium ostenfeldii, Karenia sellifornis and V. rugosum,* which produce AZAs, SPXs, GYM and PnTXs [17,60,61,62], have been previously recorded in the GuC and in the Pacific coast of Mexico. Nevertheless, Hernández-Castro did not observe these species in water samples from Bahía de La Paz, B.C.S, Mexico [48]. 

In this research, YTXs group and GYM were the more concentrated toxins in analyzed samples; however, the origin of these toxins was not recognized. Presence of epibenthic dinoflagellates of the genera *Coolia*, *Ostreopsis*, *Amphidinium and Gambierdiscus*, has been recognized in the GuC [19,48] but there is no information about their toxins in this region. Conversely, in *Coolia malayensis* from Japan, five compounds were detected and three of these were identified as analogs of YTX [63]. In 14 strains of *Coolia* spp. from Brazil, 81 compounds that include cooliatoxins, yessotoxins, ciguatoxins, maitotoxins, gambieric acids, gambierones, gambierol and gambieroxide were recently detected [64]. Putative analogues of OA and AZA were recently found in *C. malayensis* from Hong Kong, China, cultured at seven different temperatures [65]. In this study, cells of *C. malayensis* were observed in low densities in water column samples, however when samples of macroalgae, *Dictyota dichotoma,* were analyzed (data not shown), an interesting community of epibenthic dinoflagellates, including *C. malayensis*, *Amphidinium* spp. and *Prorocentrum* spp. were observed. Most likely, some analogs of YTX group detected in the bivalves’ tissues were produced by *C. malayensis*. In contrast, *K. selliformis* was not observed in samples, while low densities of *A. ostenfeldii* were found in some samples [47]. With the absence of microscopic confirmation of the presence of these species, the origin of GYMs in bivalve tissues are difficult to identify, such as mentioned by Jiang et al. 2017 [66]. Based on this information, we consider that epibenthic dinoflagellates could be the source of YTXs and CI found in the bivalves.

*Megapitaria squalida* is one of the most abundant bivalves in the Northwest of Mexico and *D. ponderosa* is recognized as a potential fishing resource [67,68,69]. In this study, both species were found in natural aggregations in sites S1, S2 and S5. Both species are sold live and usually people consume the whole organism (including viscera), raw or cooked in distinct preparations. Due to their low cost in the local market, both species are a traditional fresh seafood for residents and tourists. Pen shell *A. maura* is considered delicious seafood. Of this species, the main tissue used as food is the abductor muscle (callo). For this reason, *M. squalida* and *D. ponderosa* can represent a higher risk of human intoxication than *A. maura*.

The toxins analysis during 2015 showed low concentrations of OA and DTXs in *M. squalida* and *D. ponderosa* (<5.0 µg/Kg). Traces of AZA1 and AZA2 were detected in *A. maura* and *D. ponderosa*. PTX2 was detected in similar quantities in *D. ponderosa* and *M. squalida* in March of this year but *A. maura* from S3 had the highest concentrations of this toxin (Table 1).

The hYTX was detected during most of 2015 in *A. maura* with the highest concentrations from June to October, whereas in *D. ponderosa* and *M. squalida* this toxin was practically absent. Cyclic imines as SPXs, GYM and PnTXs were recurrent in the samples, although SPXs concentrations were lower than 4.0 µg/Kg. In samples from September and October, a higher concentration of GYM occurred in *M. squalida* than *D. ponderosa*, however, both species contained more GYM than *A. maura* (Table 1). Conversely, levels of PnTXs were ten times higher in *A. maura* than *M. squalida* and *D. ponderosa* almost all the year, except in March, when PnTXs concentrations were similar in all analyzed samples.

In 2016, between January and June, concentrations of AZAs in clams and penshell were similar, while the maximum concentration of OA was detected in June (9.6 µg/Kg) in *A. maura.* The species also exhibiting the highest concentrations of YTX in April (mainly hYTX). In contrast, in clams YTXs were practically absent except in June, when an abrupt increase was detected in *M. squalida* from S5 (Table 2). From August to November, YTXs once again were not detected in this species. Shellfish contamination from SPXs and GYM was recurrent in the analyzed samples, with a similar content of SPXs between the species from January to June but from August to November, *D. ponderosa* accumulated higher concentrations than others. The highest concentration of GYM was observed in *D. ponderosa* in January, while this CI was not detected in *M. squalida* and less than 7.0 µg/Kg was accumulated in *A. maura* in the same month. Interestingly, when both clams were sampled in the same site (S5), *D. ponderosa* accumulated more GYM than *M. squalida,* such as was documented from February to May, September and November. Conversely, both species only accumulated traces of PnTXs, while *A. maura* had the highest concentrations of this CI most part of the year (Table 2).

In 2017, the weather conditions (strong winds and waves), forced a reduction in sampling activities to five months. Nevertheless, with the reduced number of samples analyzed, a similar trend in the toxin variety and accumulation from 2015 and 2016 was observed. Again, low concentrations of OA and PTX2 were observed in *D. ponderosa*, *M. squalida and A. maura*. With respect to CI, a similar accumulation pattern was observed as in previous years. SPXs in the three species were detected in small quantities, while the concentration of GYM was higher in *D. ponderosa* than *M. squalida* and *A. maura*. In contrast, *A. maura* was the species that accumulated more PnTXs (mainly PnTXG) (Table 3). The content of PnTXs in *A. maura* suggest a higher abundance of *V. rugosum* within Ensenada de La Paz, however the presence of this species cannot be corroborated, while the presence of PnTXE and PnTXF indicates a biotransformation process in this bivalve. 

### 2.3. Intoxication Risk According to Shellfish Species

In the Gulf of California STX analogs have been described previously in *M. squalida* and *A. maura* [52,53], in contrast, data regarding LT in *M. squalida*, *D. ponderosa and A. maura* are poorly known. Toxin accumulation in bivalves is highly influenced by cell densities of HABs present in the water column and gills are the first line of contact with LT released by the cells, while the digestive gland accumulates the highest concentration of toxins [70]. For instance, a proportion of 10/1 in OA was observed between the digestive gland and other tissues in *Mytilus edulis* [71]. 

In Mexico, *M. squalida* and *D. ponderosa* are used to prepare different dishes without evisceration. The gland-stomach complex, recognized as the main repository of toxic cells, is cooked. This increases the probability of poisoning by shellfish consumption. Conversely, in *A. maura* usually only the abductor muscle (callo) is consumed, where LT content can be up to ten times lower than the concentration found in the viscera. This hypothesis was corroborated when some samples of the abductor muscle were separated from the visceral mass and mantle to be analyzed (data not presented). LC-MS/MS results showed that PnTXG was present at only 3.1 µg/Kg in the abductor muscle, whilst the visceral mass of the same specimen contained up to 29.1 µg/Kg of total LT, including 12.7 µg/Kg PnTXG. These observations fit with other reports where both lipophilic and hydrophilic toxins, in a range of bivalve species, have been quantified at significantly higher concentrations in the visceral mass in comparison with other tissues [4].

Low concentrations of DSP toxins were detected in this study, however subtle inter-species differences in toxins accumulation were observed when *D. ponderosa* and *M. squalida* were collected from the same site (S1 and S2) in Bahía de La Paz. For instance, *M. squalida* accumulated more OA than *D. ponderosa*, conversely *D. ponderosa* accumulated more SPXs and GYM than *M. squalida*. A higher accumulation of OA in *A. maura* may be related to the location of these aggregations in Ensenada de La Paz (S3 and S4), a narrow and shallow lagoon with a surface area of around of 45 Km^2^ [72]. In this region, pen shell bivalves grow at a maximum depth of 4 m. Planktonic and epibenthic dinoflagellates are resuspended in the water column by currents and turbulence where they can be filtered by mollusks. Probably, the highest concentrations of YTXs and PnTXs detected in *A. maura* were related with the presence of epibenthic dinoflagellates such as *C. malayensis* and *V. rugosum* but only *C. malayensis* was identified in this area by Hernández-Castro (2017). Although tissues of *A. maura* contained more OA, YTXs and PnTXs, the risk of intoxication by consumption of pen shell is lower than *D. ponderosa* and *M. squalida*.

### 2.4. Toxin Co-Occurrence in Bivalves

Concentrations of individual LT analogues in this study were below 160 eq OA/Kg. This evidence suggests that *M. squalida, D. ponderosa* and *A. maura* complied with the current bivalve mollusk shellfish health legislation of Mexico during the 2015–2017 sampling time in southern Bahía de La Paz. Distinct combinations of LT were accumulated in bivalve tissues, mainly GYM, SPXs, OA and DTXs in *M. squalida*; GYM, SPXs, PTXs, OA, DTXs and PnTXs in *D. ponderosa* and YTXs, PnTXs, OA, DTXs, PTXs and SPXs in *A. maura*. Due to the low concentrations of the analogs of these groups of toxins, individual effects probably cannot be detected in mouse bioassay (CD-1 mice), moreover the effects of these toxic mixtures over bivalves or in the shellfish consumers is poorly known. In 2016 the maximum concentration of combined CI (SPXs + GYM + PnTXs = 41.7 µg/Kg; SPXs + GYM = 42.1 µg/Kg), were quantified in *D. ponderosa*, while *A. maura* had the highest combination of YTXs + SPX + GYM + PnTXs = 190.8 µg/Kg), however bioassays to test this combination of LT were not performed. Although it is probable that signs of toxicity could occur, as observed by Munday et al. (2004), who perceived prostration and respiratory distress in mice, when they injected between 44.5 and 66.5 µg/Kg of GYM intraperitoneally [73].

The toxicological interactions between different toxin groups found simultaneously in shellfish consumed by humans is poorly understood. It is presumed that the combined exposure of two or more toxins will be additive with respect to dose (dose-addition) and the relative acute oral toxicities are assumed to mirror the relative acute toxicity following intraperitoneal administration [74]. For this reason, the European Food Safety Authority (EFSA) suggested to increase the research in this area, in order to gain a better understanding of the effects of the combination of different LT analogues on human health [74]. Some attempts have been reported to assess the combined effect of LT as AZA1, YTX and OA in mice, for instance when a combination of AZA1 and YTX was analyzed in mice by oral exposure, pathological changes in the intestine were similar to findings resulting from AZA1 tested alone. These intestinal damages did not increase the absorption of YTX and no clinical effects were observed with this toxin combination [75].

In another study, the combined effect of AZA1 and OA in mice was analyzed, however no acute oral toxicity, additive or synergistic effects were observed, probably due to a low degree of absorption of OA and AZA1 in the gastrointestinal tract when offered alone or in combination [76]. When an oral exposure to a combination of YTX and OA was carried out, no mortality or toxicity signs were observed, although changes in gastric and cardiac levels were revealed. When OA and YTX were dosed separately, OA induced epithelial hyperplasia of the forestomach and inflammation of its submucosa, while YTX did not induce changes [77]. Although these studies did not show additive effects when toxins were co-administered in oral doses, the combined and cumulative exposure of three or more LT (as occurred in this study) is scarcely known. Probably the effects are far more complex than simply binding to a receptor or channel; gene expression altering; changing levels of intracellular concentrations of ions; modifying the cellular metabolism; or production of cellular regulators. For this reason, effects such as antagonism, inhibition, masking, synergism or potentiation of this mixture of toxins make the prediction of their impact in shellfish populations, cultures and consequently human health difficult.

To evaluate possible synergies among OA + DTX2, OA + YTX and OA + SPX1, a previous study using a human neuroblastoma cell line was reported. Published results indicated a potentiation in OA toxicity when DTX2 was present, while the addition of YTX and SPX1 did not show effects in the neuroblastoma cell viability. When DTX2, SPX1 and OA were added together, the effect was similar to the administration of DTX2 alone [78]. The combined effects of OA, AZA1 and YTX were also studied in two human intestinal cell models. Data showed that AZA1 and YTX had a synergistic effect in Caco-2 and HIEC cell lines, mainly when high YTX concentrations were added. Mixtures of YTX and OA induced additive and antagonistic effects when one of the toxins was more concentrated. The combination of OA and AZA1 showed both additive and antagonistic effects and their toxicity in the gastrointestinal tract and epithelium decreased; suggesting that these combined toxins may result in an additive toxicity for consumers [79]. 

Overall, this is the first time that other LT, including YTXs, PTXs, AZAs, PnTXs, SPXs and GYM have been detected co-occurring with OA and DTXs in tissues of bivalves from the southern GuC, providing data about the extent of LT accumulation in these shellfish. The low concentrations of OA and analogs indicate that these mollusks comply with sanitary regulations for human consumption, however the constant presence of *D. caudata* and epibenthic dinoflagellates, such as *Prorocentrum* spp., suggest a passive accumulation of OA in shellfish tissues. In this research, the toxins with the highest concentrations during the three years of monitoring were YTXs and GYM. Studies in animal models have not shown the toxic effect of YTX combined with OA [77], the consequences of the combination of GYM with YTXs and other LT in cellular, animal models and humans, are still unknown.

During this study, it was not possible to determine a correlation between dinoflagellates cell densities in water samples and toxin concentrations in bivalve tissues. This was due to the monthly sampling strategy, that can take a long period of time to determine a relationship between toxic phytoplankton and toxin content. Bivalves showed analogs that are recognized as products of their metabolism. Moreover, phytoplankton sampling was focused on species distributed in water column, where it was possible to identify cells of epibenthic dinoflagellates, such as *Amphidinium*, *Coolia*, *Ostreopsis and Prorocentrum*. However, when some samples of macroalgae were collected, a surprisingly high abundances of cells of these dinoflagellates were observed (data not shown). During this study, the minimum surface temperature in sampling sites was 20 °C in winter, while the maximum temperature in summer was 30 °C and epibenthic dinoflagellates were observed frequently in net samples. Recently, Tester et al. (2020) hypothesized that benthic harmful algae can increase their abundances and geographic distribution associated with climate change [80]. Consequently, a regular monitoring of epibenthic dinoflagellates in this Bay is necessary, to understand the presence of LT in bivalve aggregations that sustain the shellfish fishery.

## 3. Materials and Methods 

### 3.1. Study Area and Sample Collection

In this study we conducted testing for eight groups of LTs, including CIs in homogenized tissues of wild bivalves from Bahía de La Paz (southern GuC). The Bahía de La Paz is located in the southern GuC, Mexico, between 24°06′–24°47′ N and 110°18′–110°45′ W (Figure 1). This large and deep bay is situated between the most productive coastal lagoons of the GuC [81] and is connected by an inlet to the Ensenada de La Paz, with a maximum depth of 10 m [82]. Shellfish samples representing three species (white clams, chocolate clams and pen shells) were collected from five sampling zones (Figure 1). Four individuals of each species were collected using hooka diving and transported in temperature-controlled containers on ice to the laboratory.

### 3.2. Sample Processing

Shellfish received at the laboratory were first washed to eliminate epiphytes, before each specimen was dissected to remove the muscle and viscera of the shell and pooled to be homogenized using a conventional blender. 9.0 ± 0.02 mL of MeOH grade HPLC was subsequently added to 1.0 ± 0.01 g of homogenized tissues and mixed using an Ultraturrax IKA model T18. Methanolic crude extracts were vortex mixed for 3 min and centrifuged at 4500 rpm, at 4 °C for 10 min to separate two phases. The supernatant (2.0 mL) was transferred to amber flasks and refrigerated at −20 °C. All sample extract supernatants were sent to Cefas under temperature-controlled conditions for LT analysis following the EURL SOP for the determination and quantitation of LT in live bivalve mollusks [83,84,85].

### 3.3. Sample Analysis—LC-MS/MS

Methanolic shellfish extracts were separated into two 1.0 mL sub-samples to enable analysis of both hydrolyzed and unhydrolyzed extracts. For analysis of free MeOH-extractable LT analogues, 1.0 mL extracts were filtered through a 0.2 µm nylon syringe filter and the filtrates taken for LC-MS/MS analysis. A second 1.0 mL aliquot of the raw extract was transferred into a 2 mL screw top vial for alkaline hydrolysis, involving first the addition of 125 µL of 2.5 M NaOH. After vortex mixing, the vials were heated to 76 ± 2 °C for 40 min and allowed to cool to room temperature. After this time, 125 µL of 2.5 M HCl was added to the vial and vortex mixed. The hydrolyzed extracts were then ready for LC-MS/MS analysis.

LC-MS/MS analysis was conducted using an Acquity Ultra Performance Liquid Chromatography (UPLC) with a Xevo TQ tandem mass spectrometer (Waters, Manchester, UK). A Waters BEH C18 UPLC column (50 × 2.1 mm, 1.7 µm) with a VanGuard BEH C18 (5 × 2.1 mm, 1.7 µm) guard cartridge was used for chromatographic separation. The mobile phase flow rate was 0.6 mL/min and a 5 µL injection volume employed. The mobile phases were prepared to pH 11 ± 0.2, being chosen based on the method described by Gerssen et al. 2009 [86]. Mobile phase A consisted of 2 mM ammonium bicarbonate in water with ammonium hydroxide, with mobile phase B prepared from 2 mM ammonium bicarbonate with ammonium hydroxide in acetonitrile. The gradient profile started with 75% A/25% B, holding for 0.2 min, before ramping to 50% B at 1.6 min and holding for 0.1 min. At 1.7 min, the proportion of B increased to 75% and continued to increase further to 100% at 3.0 min where it was held until 5.0 min. The proportion of B dropped back to 25% at 7.1 min, where it was held until the end of the run at 7.5 min. The total run time was 7.5 min per sample, with the column temperature set to 30 °C and sample extracts held at 10 °C. Calibration standards contained varying concentrations of each of the lipophilic toxins, available commercially as certified reference standards from the Institute of Biotoxin Metrology, National Research Council of Canada (Halifax, Canada). Certified reference standards incorporated into the calibration mixes were OA, DTX1, DTX2, AZA1-3, PTX2, YTX, homo YTX, SPX1, GYM and PnTXG. Instrumental sequences commenced with the analysis of blanks and shellfish extract samples for the system equilibration. Calibration standards were then injected at six concentration levels, running intermittently throughout the entire sequence to check for instrument response drift. Both unhydrolyzed and hydrolyzed sample extracts were analyzed in turn in between calibrants. Each target analyte was incorporated into the LC-MS/MS method with the acquisition of two Multiple Reaction Monitoring (MRM) transitions. Tandem mass spectrometer source conditions, MRM transitions and associated mass spectrometer voltages were as reported previously for this instrument [84,85]. For each analyte, one of the two MRM transitions was assigned as the primary, quantitative transition and used for quantitation of toxins in samples using the gradient calculated from the weighted (1/x) linear regression generated from the external calibration within the instrument software (Target Lynx, Waters, Manchester, UK). The second MRM for each analyte was assigned as the qualitative transition, for confirmation of analyte detection. The method has been validated at Cefas, with limits of detection and quantitation varying depending on the analyte and the shellfish species. However, the majority of analogues can be detected at concentrations < 1.0 µg/kg. 

## Figures and Tables

**Figure 1 marinedrugs-19-00099-f001:**
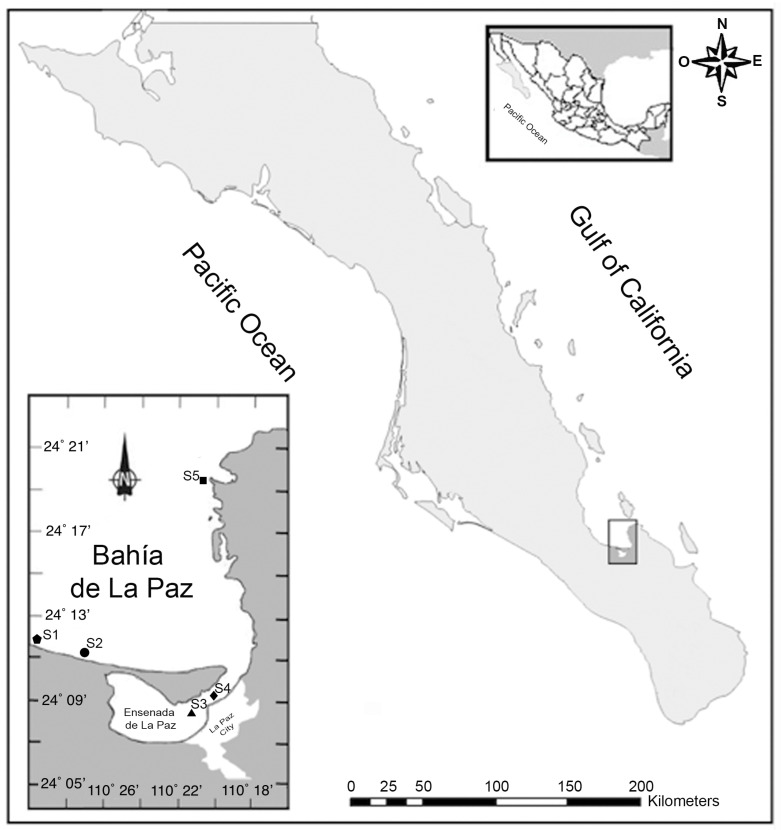
Sampling sites (S) in Bahía de La Paz (S1, S2 and S5) and Ensenada de La Paz (S3 and S4), in the southern Gulf of California (GuC).

**Figure 2 marinedrugs-19-00099-f002:**
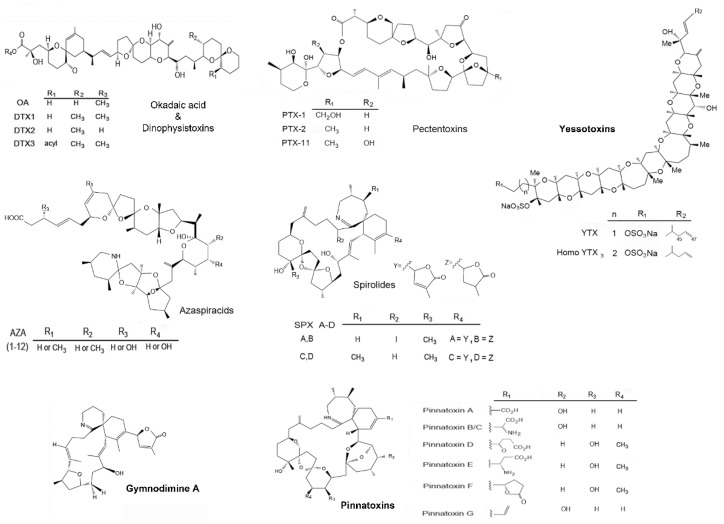
Groups of lipophilic toxins detected in three species of marine bivalves from southern Gulf of California. Names in bold indicate the groups with the highest concentrations during this study.

**Table 1 marinedrugs-19-00099-t001:** Sum for each lipophilic toxins group in bivalve mollusks from Bahía de La Paz and Ensenada de La Paz, in 2015. In bold, concentrations ≥ 10 to μg/Kg. *Dp* = *Dosinia ponderosa*; *Ms* = *Megapitaria squalida*; *Am* = *Atrina maura;* – = It was not possible to sample; * = No toxins detected; ■ = Species not present on this site.

Month.	Site	OA			DTX			PTX			YTX			AZA			SPX			GYM			PnTX		
		*Dp*	*Ms*	*Am*	*Dp*	*Ms*	*Am*	*Dp*	*Ms*	*Am*	*Dp*	*Ms*	*Am*	*Dp*	*Ms*	*Am*	*Dp*	*Ms*	*Am*	*Dp*	*Ms*	*Am*	*Dp*	*Ms*	*Am*
J	S3			0.5			*			0.05			3.3			0.02			0.3			0.3			5.3
	S4			0.5			0.05			0.05			5.3			0.01			0.1			0.1			6.8
F	S2	0.4	–		*	*		0.01	*		0.30	*		–	*		0.18	*		1.7	*		*	0.1	
	S3			0.5			*			0.1			3.9			0.02			0.2			0.22			6.32
	S4			0.8			*			0.02			5.7			0.01			0.1			0.07			**11.1**
M	S1	2.0	3.3		0.7	0.7		2.1	3.3		*	*		*	*		*	1.2		*	5.7		7.5	6.9	
	S2	0.9	–		*	–		*	–		*	–		0.9	–		*	–		*	–		0.8	–	
	S3			1.7			*			8.2			2.7			*			0.7			0.6			5.1
	S4			2.6			*			0.4			*			*			*			*			5.8
A	S1	1.7	4.2		*	*		*	*		*	*		*	*		*	*		*	*		0.9	–	
	S2	1.0	–		*	–		*	–		*	–		*	–		*	–		*	–		0.8	–	
	S3			1.5			*			1.5			*			*			2.3			1.2			7.7
	S4			2.8			*			*			*			*			*			*			**10.5**
M	S1	2.6	4.3		*	*		*	*		*	*		*	*		3.0	2.6		**10.2**	5.8		*	*	
	S2	1.3	–		*	–		*	–		*	–		*	–		2.2	–		**9.9**	–		0.8	–	
	S3			1.9			*			0.1			7.3			*			1.5			1.1			4.5
	S4			1.2			*			0.1			5.2			*			0.4			*			6.1
J	S1	3.5	3.4		0.9	–		–	–		–	–		–	–		3.3	–		**10.8**	–		*	*	
	S2	1.1	–		–	–		–	–		–	–		–	–		2.9	–		**11.7**	–		0.8	–	
	S3			4.2			*			0.4			9.5			*			0.9			0.8			4.5
	S4			3.5			*			*			**22.5**			*			0.6			0.5			6.1
J	S2	0.9	–		*	–		*	–		0.5	–		–	–		1.7	–		**24.9**	–		0.8	–	
	S3			–			*			*			5.8			*			0.9			0.9			4.9
	S4			3.5			*			*			7.3			*			0.1			–			**12.6**
A	S2	1.0	–		*	–		*	–		*	–		*	–		1.4	–		7.5	–		0.8	–	
	S3			1.7			*			*			**13.0**			*			0.5			0.8			6.6
	S4			2.6			*			*			4.1			*			0.1			*			**10.7**
S	S2	1.6	1.2		*	*		*	*		*	*		*	*		1.7	0.8		**15.5**	**26.2**		0.8	0.8	
	S3			5.0			0.9			*			**11.9**			*			0.5			**12.6**			9.4
	S4			1.9			*			*			**14.5**			*			0.1			5.6			**10.4**
	S5	–	1.8		–	0.8		–	*		–	*		–	*		–	1.3		*	**23.2**		–	2.2	
O	S2	3.6	–		0.9	–		*	–		0.5	–		–	–		1.7	–		**24.9**	–		1.7	–	
	S3			6.3			*			*			8.2			*			1.1			**11.3**			**12.3**
	S4			*			*			*			**18.9**			2.4			*			5.2			**13.1**
	S5	1.1	1.3		*	*		*	*		*	*		*	*		2.0	0.8		**19.0**	**35.4**		2.3	0.9	

**Table 2 marinedrugs-19-00099-t002:** Sum for each lipophilic toxins group in bivalve mollusks from Bahía de La Paz and Ensenada de La Paz, in 2016. In bold, concentrations ≥ 10 to μg/Kg. *Dp* = *Dosinia ponderosa*; *Ms* = *Megapitaria squalida*; *Am* = *Atrina maura;* – = It was not possible to sample; * = No toxins detected; ■ = Species not present on this site.

Month.	Site	OA			DTX			PTX			YTX			AZA			SPX			GYM			PnTX		
		*Dp*	*Ms*	*Am*	*Dp*	*Ms*	*Am*	*Dp*	*Ms*	*Am*	*Dp*	*Ms*	*Am*	*Dp*	*Ms*	*Am*	*Dp*	*Ms*	*Am*	*Dp*	*Ms*	*Am*	*Dp*	*Ms*	*Am*
J	S2	*	–		*	–		*	–		*	–		*	–		8.9	–		**33.2**	–		*	–	
	S3			*			*			0.6			*			0.9			8.9			6.8			–
	S4			*			*			*			*			*			8.4			4.7			**11.4**
	S5	*	–		*	–		*	–		*	–		*	–		**11.0**	–		**38.0**	–		0.5	–	
F	S2	*	–		*	–		*	–		*	–		*	–		9.5	–		**29.2**	–		*	–	
	S3			*			*			0.6			**11.8**			*			8.5			5.9			6.8
	S4			*			*			*			**14.0**			*			8.4			3.9			*
	S5	*	–		*	–		–	*		–	*		–	1.6		**11.5**	8.8		**30.0**	5.4		0.2	0.1	
M	S2	*	–		*	–		*	–		*	–		0.8	–		**9.9**	–		**20.3**	–		*	–	
	S3			*			*			*			**40.1**			*			9.1			6.1			6.8
	S4			*			*			*			**76.5**			*			*			*			*
	S5	*	*		*	*		*	*		*	*		0.9	1.2		**9.9**	**9.8**		**15.2**	**10.5**		0.2	0.1	
A	S2	*	–		*	–		*	–		*	–		*	–		**11.4**	–		**17.6**	–		0.5	–	
	S3			*			*			*			**29.0**			0.8			8.9			6.0			8.5
	S4			*			*			*			**88.7**			*			8.6			4.7			**11.4**
	S5	*	–		*	–		*	–		*	–		*	–		**11.3**	9.2		**23.9**	7.8		0.2	0.1	
M	S2	*	–		*	–		*	–		*	–		*	–		**10.0**	–		**12.6**	–		*	–	
	S3			*			*			1.1			**32.0**			*			8.7			4.3			3.4
	S4			*			*			*			**47.9**			*			8.2			3.5			8.1
	S5	*	–		*	–		*	–		*	–		*	1.0		**12.1**	**10.0**		**25.9**	**9.6**		0.5	0.5	
J	S2	*	–		*	–		*	–		*	–		*	–		**10.9**	–		**21.4**	–		*	–	
	S3			9.6			*			*			*			1.8			8.9			4.4			6.1
	S4			6.3			*			*			**28.1**			*			8.2			*			5.2
	S5	*	–		*	1.1		*	1.2		*	**19.2**		*	–		**11.4**	–		**17.6**	–		*	–	
A	S2	1.9	–		*	–		*	–		*	–		*	–		5.4	–		**19.5**	–		*	–	
	S3			0.8			*			*			*			*			0.5			0.9			**12.0**
	S4			1.4			*			*			5.8			*			0.2			0.9			7.9
	S5	0.3	–		*	–		*	–		*	–		*	–		1.1	–		6.6	–		*	–	
S	S2	0.6	–		*	–		0.2	–		*	–		*	–		2.4	–		**15.1**	–		*	–	
	S3			2.6			*			*			3.9			*			0.4			1.4			**12.0**
	S4			0.9			*			*			*			*			*			0.6			7.9
	S5	0.3	0.7		*	*		*	*		*	*		*	*		3.4	0.8		**11.7**	7.4		*	*	
O	S2	0.3	–		*	–		*	–		*	–		*	–		1.3	–		**20.4**	–		*	–	
	S3			2.0			*			0.1			*			*			0.9			2.9			**10.3**
	S4			1.3			*			*			**9.7**			*			*			1.0			**13.2**
	S5	*	0.1		*	0.4		*	–		*	–		*	–		1.4	0.3		5.4	**12.1**		*	–	
N	S2	1.8	–		0.5	–		*	–		*	–		*	–		1.3	–		**26.9**	–		*	–	
	S3			2.7			*			*			8.0			*			0.6			4.9			**16.7**
	S4			–			–			–			–			–			–			–			–
	S5	0.5	–		*	–		*	–		*	–		*	–		2.4	0.3		**16.1**	9.2		*	–	

**Table 3 marinedrugs-19-00099-t003:** Sum for each lipophilic toxins group in bivalve mollusks from Bahía de La Paz and Ensenada de La Paz, in 2017. In bold, concentrations ≥ 10 to μg/Kg. *Dp* = *Dosinia ponderosa*; *Ms* = *Megapitaria squalida*; *Am* = *Atrina maura;* – = It was not possible to sample; * = No toxins detected; ■ = Species not present on this site.

Month	Site	OA			PTX			YTX			SPX			GYM			PnTX		
		*Dp*	*Ms*	*Am*	*Dp*	*Ms*	*Am*	*Dp*	*Ms*	*Am*	*Dp*	*Ms*	*Am*	*Dp*	*Ms*	*Am*	*Dp*	*Ms*	*Am*
F	S2	*	–		*	–		*	–		*	–		*	–		*	–	
	S3			1.0			**10.4**			**10.0**			1.7			3.0			5.6
	S4			*			–			–			–			–			–
	S5	0.4	–		0.2	–		*	–		2.7	–		**16.3**	–		*	–	
M	S2	1.0	–		0.4	–		*	–		2.1	–		**30.2**	–		*	–	
	S3			1.3			1.5			*			1.5			4.1			8.4
	S4			0.9			2.1			6.2			0.5			1.9			2.6
	S5	0.5	–		0.1	–		*	–		2.9	–		**12.5**	–		*	–	
A	S2	0.4	–		0.6	–		*	–		1.6	–		**19.6**	–		*	–	
	S3			1.8			3.3			*			1.6			3.7			**12.8**
	S4			–			–			–			–			–			–
	S5	*	–		*	–		*	–		1.7	–		**10.7**	–		*	–	
M	S2	0.8	0.4		*	0.1		*	–		2.8	2.0		**22.1**	**16.8**		*	–	
	S3			6.4			0.4			*			1.3			1.9			**12.7**
	S4			3.8			0.6			**10.2**			0.7			2.4			8.0
	S5	0.2	0.2		*	0.1		*	–		4.9	2.7		**15.9**	**15.2**		*	–	
J	S2	0.4	–		*	–		*	–		2.0	–		**14.1**	–		–	–	
	S3			–			–			–			–			–			–
	S4			1.8			–			7.2			*			0.6			8.0
	S5	*	–		*	–		*	–		2.1	0.6		8.6	6.1		*	–	

## Data Availability

Data sharing not applicable

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
