# Peer review of "Lipophilic Toxins in Wild Bivalves from the Southern Gulf of California, Mexico"

_marinedrugs, 2021, doi:10.3390/md19020099_

Round 1

Reviewer 1 Report

Manuscript deals with lipophilic toxins determination in wild bivalves. The presented results are very attractive and their discussion is interesting.

The main drawback of this work that has to be corrected is the experimental part. Reader does not know how the results were obtained. There is no information:

  • how the MS was operated,
  • how the compound were identified
  • what standards were used.
  • how the quantitation was preformed
  • what was the LOD and LOQ
  • how the separation looks like

Some exemplary chromatogram and mass spectrum may be useful.

Think about the number of digits significant for the concentrations in the tables.

Without this information it is difficult to assess the quality of the results.

Author Response

Dear reviewer, thank you for your patience in reading our manuscript, even when several doubts arose regarding this research, that could difficult to understand during your reviewing.

Point 1:

  • how the MS was operated,
  • how the compound were identified
  • what standards were used.
  • how the quantitation was preformed
  • what was the LOD and LOQ
  • how the separation looks like

Response 1:

In order to solve your comments in these points, we made major changes in the manuscript, and a detailed description was performed about analytic method (section 3.3 Sample analysis – LC-MS/MS)

Point 2:

Some exemplary chromatogram and mass spectrum may be useful.

Response 2:

Chromatograms were included in appendix A.

Point 3:

Think about the number of digits significant for the concentrations in the tables.

Without this information it is difficult to assess the quality of the results.

Response 3:

The format of the tables and symbols were modified to highlight the toxin quantities detected in each sample, and indicating sites where: -Not was possible to sample; ✻ = No toxins detected; ^ = Species not present on this site.

We hope that with the changes made, our research can be accepted to be published in the special issue "Marine Biotoxins".

Reviewer 2 Report

Please enter your comments and suggestions for authors

Author Response

Point 1:

There are several sections missing from the material and methods. I don’t see a section on the mouse bioassay. Additionally, there doesn’t seem to be a section on the mixture analysis using the neuro-2A cells. There are no data presented for these mixture studies and it is difficult to interpret the results from the text. Was this actually carried out by the authors, or is this a review of other studies? “To evaluate possible synergies among OA+DTX2, OA+YTX and OA+SPX1 a study in human neuroblastoma cell line was carried out.” This line makes it seems like a study carried out in the current report. For instance, “The combination of OA and AZA1 showed both, additive and antagonistic effects and their toxicity in the gastrointestinal tract and epithelium decreased, suggesting that these combined toxins may result in an additive toxicity for consumers” - I don’t understand how the effects are additive and antagonistic. Also, if the toxicity decreased how would that result in additive toxicity for consumers. Am I reading this wrong?

Response 1:

First of all, I take the opportunity to thank you for your patience in reading our manuscript, even when several doubts arose regarding this research that made it difficult to understand. In this sense, we want to comment that the mouse bioassay was performed as an exploratory test, only once, ruling out DSP intoxication this bioassay made it possible to observe sublethal and lethal signs of neurotoxicity after 24 hours of intraperitoneal injection, similar to those described for fast-acting toxins. These results helped us define the toxin analyses that needed to be carried out in our samples.

We take this opportunity to clarify that in this research we only performed the detection of lipophilic toxins in mollusk extracts, the experiments that you comment to evaluate synergy, additive or antagonist effects between toxins, were examples of studies carried out by other authors, this has been clarified in the MS. We think that interactions like these could be occurring with the combination of toxins that were detected in our research, where YTX, GYM and PnTX had the highest concentrations in the bivalves analyzed, however the chronic exposure of combinations of cyclic imines with other lipophilic toxins, and their effects in animal or cell models, has been not analyzed.

Point 2:

With respect to the toxin ID and quantification work, the methods are lacking details on quantification, which standards were available, and there should be at least one LC-MS/MS spectrum included to convince readers that the identifications are valid (could be in supplementary file). If there weren’t standards, were compounds identified by LC-MS/MS fragmentation. Were MRM transitions used if the MS is capable of this?

Response 2:

In order to solve your comments in this point, we made major changes in the manuscript, and a detailed description was performed about analytic method (section 3.3 Sample analysis – LC-MS/MS).

Point 3:

Additionally, as this is a continuous data set, it would be much easier to examine the data in a graph of continuous measurements with the entire two-year period displayed. It’s difficult to go back and forth from Table 1 to Table 2. In addition, how much information can really be gleaned when there are so many instances of “not available” shellfish. For instance, in Table 2 for OA - Ms there are only 3 instances of available mollusks in the entire year period if I am reading the table right. I would encourage the authors to mine through the data more and try to visualize it in a more informative way.

Response 3:

During the elaboration of the manuscript, distinct formats of tables and graphics were explored, searching the best way to show our results. The format we present was the most simple and complete option we found in order to show our data. The format of the tables and symbols were modified to highlight the toxin quantities detected in each sample, and indicating sites where: –Not was possible to sample; ✻ = No toxins detected; ^ = Species not present on this site.

Point 4:

The authors bring up interspecific differences, but were any statistics carried out between species? Moreover, if the specimens were pooled, doesn’t that affect the ability to determine interspecific differences? What if there is one specimen that has a very high amount of toxin?

Response 4:

We agree with your comment, our results lack statistics that demonstrate interspecific differences, due that there are no replicates of each sample. When we observe the results obtained during the three years, a subtle trend of interspecific differences was detected, where one species accumulates toxins and in the others species of mollusks they are found in lower concentrations or were not detected, even if they were collected in the same place, as happened with samples of Dosinia ponderosa and Megapitaria squalida.

Our decision to analyze pooled tissues was based on the method used for the analysis of toxins in mollusk tissues for human consumption. Specimens were pooled to obtain 100 g of mixed tissues, and a subsample of two grams was used for the LC-MS/MS analysis. Under this analytic strategy, all samples had the same systematic error. We are aware that to corroborate this hypothesis an experimental design is necessary, which includes specimens of the same size, age and with no toxins in their tissues.

Point 5:

With respect to the mouse bioassay. If the specimens used were from January and February 2015, how could any symptoms exhibited by the mice be related to dinoflagellate toxins? It looks like the toxin levels are very low during that time.

Response 5:

Mouse bioassay information was removed from results and discussion section, to be included within introduction, as preliminary assay, before continue our research using an analytical method to detect lipophilic toxins. In mouse bioassay signs of DSP have been described previously by other authors, even we at the beginning of our research had focused our attention only on DSP and okadaic acid produced by Prorocentrum spp and Dinophysis spp. The low levels of individual toxins do not represent a limitation to take advantage of the bivalves as a food source, however, due to the fact that mouse bioassays were not performed in the samples where YTX, GYM and PnTX had a higher concentration, the acute effect of combined toxins is unknown for each of the samples.

Point 6:

Can low levels affect the mice in the assay? I don’t really understand the use of the mouse bioassay when toxin was going to be documented by very sensitive LC-MS/MS measurements.

Response 6:

As was described in the introduction, when DSP was discarded, signs that suggested presence of fast action toxins in the mice were observed, and LC-MS/MS was the best alternative to detect these toxins, unknown previously in this region of Mexico.

Point 7:

There were no supplementary files included in my review packet. This type of study needs some supplementary data files, especially with respect to mass spec data.

Response 7:

In addition to detailed description about LC-MS/MS analysis, chromatograms were included in appendix A.

Point 8:

Overall, I think the overall study idea is a good one, which is to carry out time-course study of dino toxins in important shellfish species to provide temporal, spatial, and species resolution with respect to toxin. This is especially important with respect to the potential of chronic intoxications. However, I think the authors need to re-envision how the study is presented in manuscript form. I would give them another attempt in Marine Drugs, but the work needs Major Revisions before it could be considered for publication.

Response 8:

We hope that with the changes made you can solve your doubts, and our research can be accepted to be published in the special issue "Marine Biotoxins".

Round 2

Reviewer 1 Report

Manuscript was corrected according to suggestions. All points are commented. In my opinion manuscript may be accepted for publication at current form.

Author Response

Dear reviewer,

we made the final corrections, and we would like to thank you again for your valuable contribution to our manuscript.

Best regards

Dr. Ignacio Leyva Valencia

Reviewer 2 Report

Overall I think the manuscript has been improved by the author revisions and I indicate minor revisions below.

Introduction line 81 change contributes to contribute

Introduction line 171 change of G. catenatum and bloom to a G. catenatum bloom

Introduction line 190 change K to Kg

Results line 316 change influence to influenced

Results line 319 - capitalize - Figure 1

Results line 321 This concentration is lower than the CI LD50 of 45 ug/Kg

Results line 323 change LC-MSMS to LC-MS/MS

Results line 493 change ephibenthic to epibenthic

Results line 663 A. maura

Results - often LTs is written where LT would be better

Methods line 1304 change firstly to first

Author Response

Dear reviewer, we made all corrections that you suggested in the final version of our manuscript, we want to thank you for your valuable contribution to improve our document.

Best regards,

Dr. Ignacio Leyva Valencia

CONACyT, IPN-CICIMAR, Mexico